# The Socio-Emotional Competencies of High School and College Students in the National Polytechnic Institute (Mexico)

**Rocío Huerta Cuervo** [1,*], **Liliana Suárez Téllez** [2], **Víctor Hugo Luna Acevedo** [3], **María Eugenia Ramírez Solís** [4], **Citlali Vela Ibarra** [5] and **Guillermina Ávila García** [4]

1  Centro de Investigaciones Económicas Administrativas y Sociales, Instituto Politécnico Nacional, Ciudad de México 11360, Mexico
2  Dirección de Formación e Innovación Educativa, Instituto Politécnico Nacional, Ciudad de México 11360, Mexico; lsuarez@ipn.mx
3  Escuela Nacional de Ciencias Bióloga, Instituto Politécnico Nacional, Ciudad de México 11360, Mexico; vhluna@ipn.mx
4  Centro de Estudios Científicos y Tecnológicos No. 14, Instituto Politécnico Nacional, Ciudad de México 11360, Mexico; meramire@ipn.mx (M.E.R.S.); gavilag@ipn.mx (G.Á.G.)
5  Escuela Superior de Comercio y Administración, Instituto Politécnico Nacional, Ciudad de México 11360, Mexico; cvelai@ipn.mx
*  Correspondence: rhuerta@ipn.mx; Tel.: +52-(55)-57296000-63115

**Abstract:** The socio-emotional competencies of students are essential for supporting their school and personal performance. This article presents the results and analysis of an experiment with students from the National Polytechnic Institute (IPN, Mexico) and the subsequent application of an instrument to assess their socio-emotional competencies. The questions that guided the research were the following: (1) What is the level of development of the socio-emotional competencies of IPN students? (2) How do variables such as sex, age, and educational level affect the degree of development of socio-emotional competencies? (3) Are the socio-emotional competencies of the study group similar to those of the control group? These questions were examined through a quantitative analysis of the results of an instrument organized into nine theoretical dimensions with 72 questions that integrate the content of socio-emotional competencies. The instrument was applied to 405 students. The results show, first, a high level of development, on average, of the students' socio-emotional competencies. Second, significant differences were found between the study group and the control group in two dimensions exclusively: empathy and autonomy. Similarly, the results show significant differences in the responses of men and women in two dimensions, regulation and pro-sociality, which, according to the analysis carried out, are related to prevailing gender stereotypes. A relevant conclusion is that the strengthening of socio-affective skills from school is essential for the comprehensive development of students. Failure to attend to them reproduces pre-existing conditions in families where poverty and violence do not allow for enriched socio-emotional environments.

**Keywords:** competencies; socio-emotional; students; evaluation; dimensions; gender

## 1. Introduction

As a result of the COVID-19 pandemic, large-scale higher education programs worldwide have faced the challenge of operating in virtual environments. Transitioning academic activities to remote modalities revealed various limitations within universities: a lack of technological infrastructure and specialized software, poor connectivity, a lack of work instruments for teachers and students, and, in general, a lack of plans to maintain continuity in emergency contexts. A critical component of this discussion is the teaching skills necessary to operate academic programs through virtual media.

When implementing teacher training, higher education organizations have, to date, focused their attention on disciplinary aspects, including the management of educational

platforms, didactic planning, the instructional design of academic platforms in virtual environments, and the preparation of relevant didactic materials for virtual environments. Nonetheless, these organizations have not focused on incorporating teacher training to identify socio-emotional problems in students and transversally incorporate content and teaching strategies for the development of socio-emotional competencies (García Retana 2012; Huerta-Cuervo and Vicario 2021). Goleman and Senge (2016) argue that:

> "If you want to educate from an integrative vision that considers socio-emotional skills as important bastions in the academic learning built by the students, you must start with the teachers." (p. 14; cited by Aristulle and Paoloni-Stente 2019)

Various authors (e.g., Gösku et al. 2021; Sirajudeen et al. 2021; Sucharitha and Amzad 2020; Sevy-Biloon 2021; Parra Castrillón et al. 2006; Trianes et al. 2002) have substantiated how stress levels, anxiety, depression, and uncertainty have increased in students since schools were deconcentrated towards homes and classes began being taught in virtual environments. Despite this, universities of different countries did not consider placing greater attention on the development of socio-emotional competencies in this period. In the case of Mexico, where 52% of the population live in poverty (CONEVAL 2021) and with one of the highest rates of social violence in the world (INEGI 2021), the importance of attending school in regard to the socio-emotional competencies of students becomes relevant. Although face-to-face interaction has limitations, in some respects (for example, sometimes there is bullying), it is irreplaceable in terms of the socialization of students. The reality is that before and during the pandemic, higher education organizations did not address their students' socio-emotional aspects, and this has happened in practically all countries.

In the following paragraphs, three aspects will be addressed: first, the origin of the concept of socio-emotional competencies; second, the results of the empirical research that was used to develop strategies for its measurement; and third, the results of research that explain the condition of students during the pandemic.

An antecedent of the concept of socio-emotional competencies appeared in 1920. Thorndike states that "social intelligence can be analyzed as a triad of abstract or academic intelligence or the ability to understand and use ideas; mechanical or visual–spatial intelligence or the ability to manipulate objects; and practical or social intelligence, which means the ability to adapt to society and social interactions" (quoted by Vaida 2016, p. 109). Thorndike states that the interest in studies on socio-emotional competencies in young adults, whether they are students or not, is relatively recent. He explains that the proliferation of such studies generates difficulties for the use and standardization of the concept, "[s]o a clarification is required... that allows successful designs and implementations of interventions for specific groups" (Vaida 2016, p. 108). Our review confirms that the concept has been used and defined in a wide variety of ways over time.

According to Bar-On (2006; cited by Ruvalcaba et al. 2019), socio-emotional competencies can be defined as a

> "[s]et of emotional, individual and interpersonal capacities that determine the ability of the individual to respond to the pressures of the environment that surrounds him." (Ruvalcaba et al. 2019, p. 89)

Although emotional intelligence derives from innate conditions (Vaida 2016), which give people who possess them essential advantages in their emotional performance, the concept of socio-emotional competencies implies that all people can acquire and develop them. The latter is possible if individuals are involved with an emotionally enriched family and educational and social activities and use explicit strategies to acquire or strengthen these socio-emotional competencies. As Goleman (2010) explains,

> "[e]motional competencies are learned skills, and having a good social awareness or being skilled at managing relationships does not guarantee mastery of the additional learning required to skillfully engage with a customer or resolve a problem." (p. 14)

Goleman (2010) systematizes what have been the most important contributions in the field of socio-emotional competencies. He explains that the "Salovey and Mayer (1990, p. 10) model sits firmly on the traditional concept of intelligence shaped by the original work on IQ". Furthermore, he argues that although IQ is relevant for measuring people's technical and cognitive abilities, it does not account for soft skills (emotional skills) that play a relevant role in people's success in specific contexts. He singles out "self-awareness, self-management, social awareness, and the ability to manage relationships" (Goleman 2010, pp. 13–14). Moreover, he associates these characteristics with the concept of emotional competencies, explaining that

> *"while emotional intelligence determines our ability to learn the rudiments of self-control and the like, emotional competency refers to our degree of mastery of these skills in a way that is reflected in the workplace."* (Goleman 2010, pp. 13–14)

Vaida (2016) establishes the difference between three related concepts: emotional quotient, emotional skills, and socio-emotional competencies.

The concept of the emotional quotient is discussed by Bar-On (1997) and cited by Vaida (2016), who suggested it to be an instrument to measure general well-being associated with the ability to understand others. Additionally, Gardner (2011) linked it to the ability to manage social interactions in explicit contexts.

According to the authors who developed these concepts (Bar-On 1997; Martínez Romero and Rojo Ramírez 2016; Goleman 1995; cited by Vaida 2016), emotional skills "are specific qualities necessary to perform a given task" (McFall 1982; cited by Vaida 2016), and these qualities moderate emotional traits and competencies.

Building on previous studies, Seal and Andrews-Brown (2010) suggested the concept of socio-emotional competencies (Vaida 2016). In contrast to emotional intelligence, which is related to inherited characteristics, socio-emotional competencies can be strengthened by experience in emotionally rich contexts of coexistence as well as by specific and professional interventions aimed at enhancing them. Seal and Andrews-Brown (2010) identified four dimensions of socio-emotional competencies: self-awareness, respect for others, connecting with others, and having a clear orientation to change (cited by Vaida 2016).

In a documentary study carried out by Alvarez (2020) on the advantages of socio-emotional education in schools, the author concludes:

> *"Among the purposes of socio-emotional education are the prevention of social problems whose prevalence is increasing, such as violence, addiction . . . Another of its purposes is cognitive and has to do with the development of skills and abilities to achieve outstanding performance, enhance creativity and achieve effective management of stress and pressure in the workplace . . . "* (pp. 12–13)

Along the same lines, Goleman (2010) points out that, to date, access to socio-emotional education has essentially been limited to the most economically advantaged segments of the population and not the majority of children and young people. This perpetuates a vicious cycle in which the poor do not find opportunities to fully develop, and their opportunities for humane and healthy conditions of existence are limited by their societies (p. 17).

Finally, some of the reviewed studies systematize the concept of socio-emotional competencies, picking out specific relevant elements. Boyatzis (2009; cited by Vaida 2016, p. 109) defines socio-emotional competencies as the "set of interrelated behaviors that are organized according to an intention and that lead to success". Oberst et al. (2009) define this concept as "the description of learning outcomes, that is, what a person knows or can demonstrate to have learned, after a learning process". Moreover, Bizquerra, Alzina and Pérez Escoda define it as "the ability to adequately mobilize the set of diverse knowledge, skills and attitudes with a certain level of quality and effectiveness (Mikulic et al. 2015, p. 22)." These three definitions emphasize that competencies are developed from training processes (learning), coexistence, and teamwork.

This section highlights research carried out before the pandemic. The purpose of this is to identify the level of development of socio-emotional skills in students, especially in high school and higher education.

Ruvalcaba et al. (2019, pp. 91–92) conducted an "observational, explanatory and cross-sectional" study on 840 students aged 12 to 17 years from the Guadalajara metropolitan area. They adapted the instrument built by Bar-On (2006), which consists of 48 items rated on a Likert-type scale. These items were grouped into six dimensions: intrapersonal skills, interpersonal skills, anger management, adaptability, positive emotions and optimism, and self-concept. They also identified what they call "components of resilience" (personal competencies, social competency, family cohesion, social resources, and goal orientation), finding "that adaptability and the generation of a positive mood facilitate personal competency associated with resilience" (p. 95).

Aristulle and Paoloni-Stente (2019) carried out an investigation using a survey to identify the self-perceptions of the socio-emotional competencies of 53 students who were training as teachers of initial and primary education. The instrument consisted of 33 items:

> "For each skill or item, each student had to decide on three aspects: (1) the degree to which they perceive they have developed the mentioned skill by marking the corresponding option with a cross or check mark; (2) if the others (peers, teachers, parents, classmates, etc.) consider that they have developed said ability; (3) if the skill in question is important to their performance at the higher level." (p. 13)

In the interpersonal component, the students had an average of 54.7%, which indicates that they perceived significant limitations when interacting with others, understanding others, doing group work, and exhibiting leadership qualities. As for the intrapersonal competencies, the average was 54.1%, which means they felt that they had not sufficiently developed the ability to become aware of their feelings when they experienced them or expressed them. The results were similar for stress management and the ability to adapt and make adjustments according to changing environments. Only in the mood dimension was the average higher than 80%.

Rendón Uribe (2015) sought to identify the relationships between teaching strategies and the development of socio-emotional competencies. The author presented the results of an empirical study through mixed-methods research that integrated semi-structured interviews, a questionnaire with open questions, an observation guide, a questionnaire to identify teaching strategies (28 items), and another questionnaire to identify socio-emotional competencies in students (72 items). The author interprets his findings as follows:

> "Although the students obtained a high average score in the socio-emotional competencies test, there are coexistence and socio-emotional problems in competencies such as: self-efficacy, self-regulation, self-control, problem solving, social skills and empathy. Teaching styles have a direct relationship with the education of students' socio-emotional competencies to the extent that they allow or not environments conducive to dialogue, conflict management and the strengthening of emotional and social competencies, which in turn." (p. 252)

Mikulic et al. (2015) sought to validate the Inventory of Socio-Emotional Competencies (ICSE) for adults as an instrument for identifying and measuring socio-emotional competencies in young adults. This validation aimed to "[become] a valuable tool, both to be used in research activities and intervention and prevention programs" (p. 312). Building on the contributions of Hogan (2004; cited by Mikulic et al. 2015) on the research strategy for the construction of diagnostic tests, the authors followed the following steps to validate the instrument: (1) Carrying out a bibliographic review and operationalization of the construct, (2) preparing the items, (3) conducting an analysis by expert judgment, and (4) performing a pilot test and analysis of the psychometric properties.

Four hundred and forty-six participants took part in the pilot test of the instrument, and the Kaiser–Meyer–Olkin (KMO) index was used to refine the instrument based on those results, leaving it with 72 items. The KMO test indicates the proportion of variance

that the analyzed variables have in common. Five thousand and nine people from Buenos Aires and its suburbs participated in applying the refined instrument. The authors verified the normality in the distribution of the results; only eight items had inadequate values. According to the authors, "[t]he Kaiser–Mayer–Olkin measure of sample adequacy (0.843) and the Bartlett sphericity test indicated the feasibility of performing factor analysis" (p. 314). A result that was considered relevant was that only nine factors explained 42.57% of the total variation, which has a bearing on the identification of educational intervention policies within the curriculum and instructional design. These nine factors "correspond to the dimensions of self-efficacy, optimism, assertiveness, emotional expression, emotional awareness, empathy, emotional regulation, pro-sociality and autonomy" (p. 314).

Socio-emotional competencies can function as protective factors that facilitate a better adaptation of the subject to the context. It has also been observed that when well developed, these competencies favor learning processes, problem-solving, obtaining and maintaining a job and work, and professional performance (Bisquerra Alzina and Escoda 2007; cited by Mikulic et al. 2015, p. 330).

After conducting the review summarized above, the authors of this project decided to apply the Mikulic et al. (2015) instrument because it integrates the variables associated with each of the dimensions considered by the socio-emotional empirical studies.

Regarding research related to work in virtual environments, an experiment conducted by Mather and Sarkans (2018) at Centennial College in Toronto, Canada, consisted of teaching the same class to two groups of students, one working in virtual environments and the other in person. The authors found that while students who attended class virtually valued the increased flexibility and ability to attend classes without commuting, they expressed problems with the use of technology and providing feedback on their results from the teachers promptly. They also noted a lack of clarity in communication with their teachers and classmates, believed that the work was not distributed equally among the group members, and had limitations in carrying out teamwork. Although the students who took their class in person also stated that their main challenge was teamwork, they highly valued the socialization they were able to engage in with their classmates and "praised their teachers for the clarity of instruction, variety of instructional strategies and genuine interest in student learning" (p. 70). In both groups, 70% stated that their results were excellent.

This study is significant for our purposes because it constitutes evidence from before the pandemic that one of the weaknesses of virtual teaching and learning environments is a lack of attention to socialization processes, which are essential in the training and mental health of students. As outlined below, these unfavorable repercussions of virtual learning have also been observed during the pandemic.

Herold and Chen (2021) presented the results of a survey carried out on the directors of operations of the virtual environments of universities in the United States. In this survey, respondents reported that, before the pandemic, 51% of the students were very satisfied with their classes. This percentage dropped to 19% during the pandemic. According to principals, only 32% of students had "moderately positive" attitudes to their online classes, and just 17% said they were very satisfied. "In particular, students consistently reported increased stress and decreased ability to concentrate . . . barriers to collaborating with peers, difficulties paying attention, staying focused, and staying motivated" (p. 322). A total of 48% and 38% of the students reported moderate to high levels of depression and anxiety, respectively, and 18% said they had thoughts of self-harm or suicide. A total of 38% showed moderate to high levels of anxiety.

Herold and Chen (2021) also surveyed 168 psychology students and found that "[t]he conditions of the students changed with the pandemic. Although 47% lived with their parents before the pandemic, after the spring of 2020, the percentage grew to 81%". Of those surveyed, 75% had to continue working outside their home; the rest worked as "maids, caretakers, and supermarket buyers" in their own homes, which led to an increase in stress and anxiety and decreased concentration in their classes.

According to Gösku et al. (2021), stress, anxiety, depression, and uncertainty intolerance were present in Turkish higher education students during the pandemic. The authors found that "these variables were negatively correlated with distance learning motivation and distance learning attendance frequency" (p. 2). They identified that students who were able to take synchronous and asynchronous classes simultaneously enjoyed the courses more.

Sirajudeen et al. (2021) found that health career students from Saudi Arabia experienced distress in their learning processes and had discipline problems such as difficulty concentrating. Family quarantine conditions due to illness, anxiety due to academic delays, and loneliness negatively affected the students, which had negative repercussions on their performance (p. 778). Most of the students who participated in their research (63.4%) had no previous experience working in virtual environments.

Sevy-Biloon (2021) surveyed the perceptions of 69 students training to become English teachers on their preferences for studying in virtual environments or in person during the pandemic. The results showed that 47 students preferred face-to-face classes because their questions could be answered more efficiently and they could interact with their classmates. These students indicated not being affected by the distractions that working from home generates. The author explains "how insights can help understand a person's situation, which can help the teacher create a better learning environment for students" (p. 29). This is important because perceptions affect people's well-being and, therefore, their ability to engage in and be stimulated by teaching and learning processes.

Although the studies reviewed above show differences in the level of the students' digital skills (in terms of their ability to handle information and communication technologies) according to the country and university where the study was carried out, most of the investigations concluded that the students presented severe problems of stress, depression, anguish, and uncertainty. These issues were a consequence of the pandemic condition itself and the modification of spaces and modalities for learning. These changes hurt students' ability to concentrate and, in some cases, their level of achievement. The communication and interaction between teachers and students in face-to-face learning environments are what students miss most significantly in virtual environments.

The central objective of this article is to analyze the level of development of the socio-emotional skills of IPN high school students and other high school students based on the results of the application of an instrument to evaluate them (Appendix A). In parallel to this, this study aims to identify the differences in the development of these skills derived from the sex and educational level of the participating students. A specific objective of the research is to know the impact of the sociability activities carried out with the study group during the semester prior to applying the evaluation instrument. We aimed to identify if these activities made any difference in the development of the socio-emotional competencies of the two groups. The hypotheses were the following:

**Hypothesis 1 (H1).** *IPN students have a low level of socio-emotional competencies.*

**Hypothesis 2 (H2).** *The level of socio-emotional competencies is significantly different between the study group and the control group.*

**Hypothesis 3 (H3).** *There are significant differences in the socio-emotional competencies (dependent variable) of the students based on age, sex, and educational level.*

A reflection derived from the theoretical review was that systematic and permanent processes are required throughout the different educational levels to develop the socio-affective competencies of young people. Students' attitudes and the attitudes of people, in general, are built throughout their lives. Deconstructing the limitations in developing these skills cannot result from short-term experiments but rather from continuous and well-organized interventions by the school. An experiment carried out for a single semester can hardly make a difference in students' socio-emotional skills, even more so in contexts such as Mexico, in which young people are constantly exposed to unfavorable family social

environments (World Health Organization 2002). It is unavoidable that schools have an obligation to attend to these realities that limit the integral formation of students. Moreover, these realities limit the conditions for students' future insertion into the labor market, their lives as citizens, and their performance within a family and community.

## 2. Materials and Methods

### 2.1. Analysis Methodology. Dimensions of Socio-Emotional Competencies

The combined population of the upper-middle and upper levels of the IPN was 202,576 students, as appears in Table 1. The sample included 405 students who participated in the study. Although we might expect the sample to be representative of the total student population, given that it was not randomized or stratified, this cannot be affirmed.

**Table 1.** Total population of the IPN.

|  | IPN Student Population |
| --- | --- |
| Medium superior level | 74,509 |
| Upper level | 128,067 |
| Total | 202,576 |

Note: Data taken from the National Polytechnic Institute (2020) Statistical Yearbook.

Two groups were identified for the experiment. The proposed socialization activities were carried out in the study group and not in the control group.

The invitation to participate in the experiment was extended through professors who participate in the IPN teaching network, the Rethinking Seminar Network (Huerta-Cuervo et al. 2020; Ruiz et al. 2020), where professors from the upper-middle and higher levels participate. There were four teachers who agreed to participate in the experiment, including two teachers of the higher level and two teachers of the upper-middle level. Two teachers were assigned to the study group and two to the control group. The instrument to assess socio-emotional competencies was applied at the end of the semester.

The study group consisted of 237 students, of which 136 were from the upper secondary level and 101 were from the higher level. The control group consisted of 168 students, of whom 133 were from the upper secondary level and 35 were from the higher level. The characteristics of the participants in the study and control groups are outlined in Table 2.

**Table 2.** Characteristics of the participants of the study and control groups.

|  | Female | Male | Other | High School Students | Higher-Level Students | Total |
| --- | --- | --- | --- | --- | --- | --- |
| Study group | 111 | 125 | one | 135 | 102 | 237 |
| Control group | 63 | 105 | - | 134 | 3.4 | 168 |
| Total | 174 | 230 | one | 269 | 136 | 405 |

Note: Data taken from the applied questionnaire for this research (Huerta-Cuervo and Vicario 2021).

We were unable to apply the instrument proportionally to the total number of students for each educational level because the invitation to participate was extended through a specific group of teachers. The teachers who accepted and who worked at the upper-middle level had larger groups than those who taught at the higher level. In addition to the 72 questions derived from the instrument proposed by Mikulic et al. (2015), the instrument used in this research included four identification questions (group, sex, educational level, and age).

The two groups of participants were clearly identified. High school students, in general, had a median age of 17 years, and higher-level students had a median age of 22. There was also a difference between the average ages of men and women. The average age of the women was 19.5 years and that of the men was 18 years.

### 2.2. Analysis Methodology

In our review in the previous section, we observe that there is a great deal of agreement in the dimensions that the various authors we cited identify as components of socio-emotional competencies, although not all of them consider the same number of dimensions. The instrument proposed by Mikulic et al. (2015) incorporates nine dimensions: awareness, regulation, empathy, expression, autonomy, self-efficacy, pro-sociality, assertiveness, and optimism. Each of these dimensions is measured by a set of items, such as those shown in Appendix A. The instrument integrates items that capture the content of the concept of socio-emotional competencies derived from the review carried out. The meaning of each dimension appears in Table 3.

**Table 3.** Dimensions of the analysis.

| Instrument for the Assessment of Socio-Emotional Competencies Theoretical Dimensions | |
|---|---|
| Awareness | To understand emotions. Being able to differentiate them in a significant way. Specifying what is known as emotional signals and the expressions through which they are manifested in people. |
| Regulation | Self-regulating activities. Knowing oneself as a basis for decision-making processes. |
| Empathy | It is a way of understanding the emotional state of another person and being understanding of their situation. |
| Expression | This dimension incorporates communication; it can be verbal or non-verbal; it is expressed consciously and establishes a bridge on which thoughts travel in order to be understood. |
| Self-efficacy | Indicates the commitment that the person has to himself, involving the ability to achieve the objectives that are self-proposed as well as the ability to design strategies. |
| Pro-sociality | It is expressed among people in the form of actions that are performed for the benefit of others. |
| Assertiveness | This implies an appropriate form of expression to express positions and defend their own and others' rights. |
| Optimism | Looking for the positive side of situations, even under disadvantageous conditions, thus managing to maintain a positive attitude. |
| Autonomy | The ability to make conscious decisions by oneself. |

Note: (Mikulic et al. 2015, p. 318).

Thus, the research strategy defined was quantitative. The central objective of the research was to measure the socio-emotional competencies of the students with the application and analysis of the instrument. Five levels or ranges were identified in the development of socio-emotional competencies: very high, from 4.1 to 5; high, from 3.1 to 4; average, from 2.1 to 3; low, from 1.1 to 2; and very low, from 0 to 1.

Prior to the application of the defined instrument, an experiment was designed and carried out. The goal was to incorporate socialization activities in virtual environments for the students of the study group throughout a semester in order to verify if these types of exercises impact their behavior. It consisted of recovering the experience of Huerta-Cuervo and Vicario (2021). Socialization activities were designed for students in virtual environments in such a way that, prior to their classes, young people had time to choose either from a topic proposed by the teacher or by themselves (Appendix B). During ten classes, the students had a period of 20 min for these activities. Small groups were formed inside the virtual room to promote a better rapprochement and dialogue. The topics suggested for socialization were diverse, such as discussing a movie, a painting, a sporting event, or an article in the newspapers. It should be noted that in various sessions, the

same young people introduced the topic of the talks. The objective of these activities was to partially replace, within the limitations of virtual environments, the socialization areas provided by in-person schools (cafeteria, corridors, and living areas) to give the students the opportunity to live and talk among themselves. The teachers were not present during these dialogues.

For the analysis of the answers obtained, the statistical packages InfoStat and SPSS were used (see Supplementary Materials). Additionally, the following steps were taken:

1. To measure the socio-emotional competencies of the students, the instrument was applied to all 405 students at the end of the semester.
2. The questionnaire was applied in the format of Google Docs. All responses were converted from the Likert scale to a numerical scale, where a score of five represented the more desirable or positive end of the scale and one the least desirable. For example, the statement "it is difficult for me to control my emotions" would be a score of one on the scale. Each response column was individually transformed to ensure that they were comparable.
3. The responses were grouped for each dimension, and the average per student was obtained for each of the nine dimensions that made up the questionnaire. Thus, a table of 405 rows by 13 columns was obtained, 4 with the identification data (study or control group, sex, age, and educational level) and 9 with the results of the dimensions considered.
4. Cronbach's alpha internal reliability index was obtained, which was 0.9617.
5. A joint analysis was performed to identify statistically significant differences between the two groups; later, the results were also contrasted using the variables of age, sex, educational level, and the type of group.
6. In order to know which of the different variables are more important in regard to the socio-emotional competencies, we obtained the factorial analysis through principal component analysis with varimax rotation.

## 3. Results

First, each of the results was evaluated in terms of the dimensions of the analysis. The students obtained, on average, a high score in the development of socio-affective competencies in all dimensions (Table 4). Optimism and pro-sociality are the dimensions that obtained the highest scores and empathy and regulation the lowest, on average. With the above, hypothesis H1 was rejected. In this sense, we can affirm that despite the problems caused by the pandemic, the IPN students, in general, maintained a positive attitude towards the events and that the pandemic did not prevent them from keeping in touch with their classmates.

**Table 4.** Averages obtained in each dimension of the socio-emotional competencies.

| Assertiveness | Self-Efficacy | Autonomy | Awareness | Empathy | Expression | Optimism | Pro-Sociality | Regulation |
|---|---|---|---|---|---|---|---|---|
| 3.32 | 3.27 | 3.44 | 3.41 | 3.07 | 3.17 | 3.7 | 3.62 | 3.13 |

Note: Data taken from the instrument of evaluation (2021).

The fact that empathy and regulation were the dimensions with the lowest scores (3.07 and 3.13, respectively) suggests that the students had difficulties in understanding and putting themselves in the place of others and points to the limitations in the students' ability to organize and act according to clearly defined objectives.

The H1 hypothesis was rejected using the means of each of the nine dimensions (Table 4) according to the established typology.

Although the results are more positive than the research group expected at the beginning of the study, they show many areas that can be addressed in the training process. This is especially true given that 23% of students showed an average score of less than three, which indicates a medium or low level in the development of their socio-emotional competencies. In Appendix A are the items that make up each dimension.

When the analysis was performed for each of the dimensions, no significant differences between the two groups were found in the following dimensions: assertiveness, self-efficacy, conscientiousness, expression, optimism, pro-sociality, and regulation. There were relevant differences only in the dimensions of empathy and autonomy (Table 5). In the case of the empathy dimension, the averages were 2.65 for the control group and 3.37 for the study group. In the autonomy dimension, the averages were 3.54 for the control group and 3.37 for the study group.

**Table 5.** Hypothesis test of the autonomy group.

| | | | | Kruskal–Wallis Test | | | | |
| --- | --- | --- | --- | --- | --- | --- | --- | --- |
| **Variable** | **Type** | **Group** | **N** | **Mean** | **SD** | **Median** | **H** | ***p*** |
| Autonomy | 0 | Control | 168 | 3.54 | 0.72 | 3.6 | 6.17 | <0.0126 |
| Autonomy | 1 | Study | 236 | 3.337 | 0.64 | 3.4 | | |
| Empathy | 0 | Control | 168 | 2.65 | 0.63 | 2.6 | 102.79 | <0.0001 |
| Empathy | 1 | Study | 237 | 3.37 | 0.6 | 3.4 | | |

Note: Data taken from the Kruskal–Wallis hypotheses test. INFOSTAT.

H2 was partially accepted. A Kruskal–Wallis one-way analysis of variation for non-parametric samples was carried out (without presupposing any type of data distribution) to evaluate the means differences of the nine dimensions (Table 5).

It can be inferred that in the case of empathy, the socialization exercises carried out prior to the start of classes during the semester helped to generate rapprochement and trust among the members of the study group, which was reflected in the result, where the difference is not only significant but very large.

Regarding the differences in the autonomy dimension (which were not expected), in which the result was higher in the control group, it is likely that the idea of feeling watched or observed by the teachers influenced the study group and, therefore, they manifested less autonomy than the control group. In the spaces for socialization, the importance of paying attention to what the members of the group have expressed and, based on that, building arguments and having an opinion was commented on. Regarding the question "I depend on others to make decisions", the score was higher in the study group. Hypothesis H2 was partially accepted.

One relevant finding is related to the sex of the participants. This variable affects two dimensions of the analysis carried out, which are regulation and pro-sociality (Table 6). Although women showed a higher score in pro-sociality (mean of 3.72), men showed higher punctuation regulation (3.25). Similar to the experiment carried out by Mikulic et al. (2015), the competency of pro-sociality was more developed in women. H3 was partially accepted because there were significant differences in three socio-emotional competencies (pro-sociality, regulation, and empathy) of the students (dependent variable), taken as a single group according to their sex and educational level (Table 6).

**Table 6.** Hypothesis test of pro-sociality and regulation according to sex.

| **Variable** | **Sex** | **N** | **Mean** | **SD** | **Median** | **H** | ***p*** |
| --- | --- | --- | --- | --- | --- | --- | --- |
| Pro-sociality | 0 | 230 | 3.54 | 0.57 | 3.5 | 7.96 | <0.0046 |
| Pro-sociality | 1 | 173 | 3.72 | 0.62 | 3.67 | | |
| Regulation | 0 | 230 | 3.25 | 0.76 | 3.29 | 8.02 | |
| Regulation | 1 | 173 | 2.98 | 0.77 | 3 | | <0.0045 |

Note: Data taken from the Kruskal–Wallis test of the hypotheses.

Gender stereotypes, as a sociocultural construction that limits the possibilities of the comprehensive development of the sexes, must also be addressed in the educational process. The variables that are considered in the regulation dimension appear in Table 7.

**Table 7.** Variables that comprise the regulation dimension.

| I find it hard to act calm when something makes me very nervous | I find it difficult to control my emotions | I lose control when something makes me angry | When I'm angry, the worst of me comes out | When faced with a problem, I find it hard to think clearly | When someone offends me, I am able to stay calm | I can handle my emotions |
| --- | --- | --- | --- | --- | --- | --- |

Note: Data taken from Mikulic et al. (2015).

Traditionally, women have been characterized as beings without the ability to control their emotions. Such characterizations do not stem from innate or biological conditions but rather from cultural conditions that restrict the life and behavior of family members and society as well as the types of tasks that have been historically assigned to people of different genders. The same is true in the case of the difference in pro-sociality. Lack of pro-sociality is not a natural condition in men either but has been defined by the imposed social division of labor and the values associated with "being a man", in which "not complaining", "not showing weakness", and so on have been conventionalized and internalized (Huerta-Cuervo et al. 2020).

We expect that promoting strategies for students to strengthen their regulatory capacities will improve their professional, family, and community performance in the future. The male students, despite having higher scores than the female students, were not at an optimal condition either (average of 3.25 out of 5), which, along with the elements discussed below, provides the basis for a training proposal in terms of socio-emotional competencies.

Another statistically significant difference that was identified between the groups came from their classification according to educational level. Both pro-sociality and empathy were correlated with students' education level. In both variables, higher-level students had higher mean scores, 3.71 for pro-sociality and 3.29 for empathy, compared to the lower-level students, who had mean scores of 3.57 for pro-sociality and 2.96 for empathy. To test the significance of the potential positive relationship between empathy and pro-sociality and age, the respective Kruskal–Wallis hypothesis tests were run. Only empathy showed significant differences between age groups. The lowest averages were found not among the youngest students but in the age group from 18 to 20 years old, with a mean of 2.78, compared to a mean of 3.22 for students between 15 and 17 years old and 3.46 for those over 21 years old. In order to know which of the different variables are more significant in regard to socio-emotional competencies, in the next table, we show the result of principal component analysis. Factorial and principal component analyses can help find interrelationships between variables and reduce the variables to highlight the most relevant ones. In this case, these techniques were used for confirmatory purposes in order to support the results previously obtained by Mikulic et al. (2015). Previously in this exercise, the data were normalized with the Z-score technique. The results of the principal component analysis indicate the weight of each eigenvalue in the explanation of the variance in relation to the total. The results of the exercise carried out highlight three eigenvalues, which, together, explain 68.56% of the total variance (Table 8). To identify how many factors we were left with, the Kaiser criterion was used, with eigenvalues equal to or greater than 1.

**Table 8.** Explained variance according to the components.

| Components | Eigenvalue | Proportion | SE_Prop | Cumulative | SE_Cum | Bias |
|---|---|---|---|---|---|---|
| Component 1 | 3.9447 | 0.4383 | 0.0186 | 0.4383 | 0.0186 | 0.0158 |
| Component 2 | 1.2062 | 0.1340 | 0.0093 | 0.5723 | 0.0155 | 0.0252 |
| Component 3 | 1.0194 | 0.1133 | 0.0080 | 0.6856 | 0.0123 | −0.0034 |
| Component 4 | 0.6886 | 0.0765 | 0.0056 | 0.7621 | 0.0100 | 0.0834 |
| Component 5 | 0.6751 | 0.0750 | 0.0055 | 0.8371 | 0.0075 | −0.0857 |
| Component 6 | 0.5067 | 0.0563 | 0.0042 | 0.8934 | 0.0054 | −0.0072 |
| Component 7 | 0.4079 | 0.0453 | 0.0034 | 0.9387 | 0.0036 | −0.0084 |
| Component 8 | 0.3392 | 0.0377 | 0.0029 | 0.9764 | 0.0018 | −0.0128 |
| Component 9 | 0.2123 | 0.0236 | 0.0018 | 1.0000 | 0.0000 | −0.0068 |

Note: Data taken from Stata.

The three components that explain the variance of the dependent variables in a better way were denominated as follows:

Component 1 is the personal component; component 2 is the social component; component 3 is the link with the other components because of the elements that are highlighted in each one of them (Table 9).

**Table 9.** Principal components (eigenvectors).

| | Rotated Components | | | |
|---|---|---|---|---|
| Variable | Personal Comp1 | Social Comp2 | Link with Others Comp3 | Unexplained |
| Assertiveness | 0.3827 | −0.1110 | 0.2609 | 0.3271 |
| Auto−efficacy | 0.4639 | −0.0563 | 0.0587 | 0.2019 |
| Autonomy | 0.0430 | 0.0666 | 0.8243 | 0.1689 |
| Awareness | 0.4246 | −0.0500 | −0.0154 | 0.3604 |
| Empathy | −0.0025 | 0.7168 | −0.2416 | 0.2699 |
| Expression | 0.4283 | 0.0489 | −0.1579 | 0.3223 |
| Optimism | 0.3820 | 0.0828 | −0.03017 | 0.4100 |
| Pro−sociality | 0.0056 | 0.6680 | 0.2637 | 0.3209 |
| Regulation | 0.3560 | 0.0912 | 0.096 | 0.4484 |

Note: Data taken from Stata.

## 4. Discussion

Socio-emotional competencies are a substantial component in the performance and behavior of people; therefore, their development must be a priority in educational centers. These competencies explain how students behave and respond to situations in a variety of contexts; the competencies impact their academic success and their ability to cope with the challenges of their environments (Goleman 2010).

Individuals' success does not depend only on the knowledge and instrumental skills they possess or the social group in which they were born and grew up but on the set of skills they manage to demonstrate. Hence, a person's socio-emotional competencies can play a crucial role in the achievement of their personal objectives (or failure to accomplish them). According to Goleman (2010), education provides opportunities that can only be expanded if students' socio-emotional competencies are developed since they will be stronger and more confident.

Among their responsibilities as equalizers of opportunities, schools need to train and develop students' socio-emotional competencies. Failure to do so will limit the ability of students to reach their goals. As Gardner (2011) emphasized, managing social interactions is a crucial ability to have in academic, professional, and personal settings. In societies such as the Mexican society, with poverty and inequality prevalent in significant segments of society, failing to train students in socio-emotional skills leaves them vulnerable to growing up in circles dominated by gender stereotypes, violence, and ignorance of these issues, as well as the material limitations of family life.

According to Alvarez (2020), emotions can serve as an effective intervention in the teaching–learning process because they can impact teacher–student interaction. Our study findings suggest that the series of previously formulated activities and the creation of a climate of trust and respect contributed significantly to the development of the proposal presented by the students, i.e., they facilitated oral participation with arguments specific to a topic. The students stressed that this activity was important because it allowed them to reflect on matters other than family problems or the same learning module. They also appreciated the chance to feel closer to each other again, although they mentioned that it was nothing compared to the physical contact that they were used to. Finally, students cited that sociability activities were a good way to start classes since they thought these activities allowed them to perform better as they felt less overwhelmed by information. Mather and Sarkans (2018) explain how students can feel comfortable with the synchronous and asynchronous modes of instruction. However, online courses lack both academic and personal elements, which hinder this personal contact. Schools have not paid attention to the socio-emotional conditions of students, not even in the most basic ways, which may have limited the results of two years of virtual schooling. This mostly affected the percentage of students displaying low socio-emotional competencies.

Along with the results of Mikulic et al. (2015), this study confirms that the application of the instrument is valid and reliable for the evaluation of socio-emotional competencies. Additionally, the results can potentially be used in later analyses of hybrid education models.

## 5. Conclusions

In order to be effective, educational centers should take on the responsibility of developing the socio-emotional skills of their students as one attribute that can positively affect not only their school performance but also their personal and group performance. Social inequalities can be reduced by giving students the tools to reflect on their behavior and, equally, to improve their responses to events in diverse contexts. By utilizing the instrument and implementing the experiment, it was possible to:

1. Identify the importance of sociability activities among students to develop socio-emotional skills in virtual environments.
2. Determine that although the "virtual sociability" activities that were carried out with the study group during the semester favored the development of empathy competence, it is important to underline that in order to develop all the dimensions of socio-emotional competencies to a high degree, continuous and systematic processes are necessary throughout the education cycle. Emotionally re-educating young people who have not participated in emotionally enriching environments or fully developed the competencies they already possess cannot be the product of partial care processes.
3. Propose to deconstruct gender stereotypes that limit both men and women from utilizing their strengths, abilities, and competencies through reflection strategies with students.
4. Identify the three components that most influenced students' behavior and socio-emotional skills.
5. Observe that the average score obtained in each of the nine constructed dimensions was greater than three but less than four, indicating that IPN students, in general, are at a high and medium level of development in terms of socio-emotional competencies; only 25% showed medium and low levels in these competencies. This finding supports the importance of explicitly strengthening socio-emotional competency training processes (Rendón Uribe 2015; Goleman 2010).

As this was not the study's objective, it was not possible to compare this result with the participants' academic performance. Nonetheless, this is an aspect that will require further study as it can shed light on intervention strategies for students at greater academic and social disadvantage.

In the IPN, as well as in education organizations at different levels, it is essential to conduct broader exercises to assess the level of socio-emotional development in students.

In contexts where poverty is high, social and intrafamily violence is prominent, and gender stereotypes are prevalent, schools must be devoted to the development of their students. In particular, focus should be placed on building socio-emotional competencies, which are the basis for student success, not just at school but also in the family and social spheres.

We identified that the skills of self-efficacy, empathy, pro-sociality, and autonomy are crucial to students (parameters higher than 0.4).

Instruments such as the one used in this study are valuable in measuring socio-emotional skills (Mikulic et al. 2015). Students were able to discuss non-subject topics in a new way, and far from feeling overwhelmed, they were able to express their emotions. When implementing this type of activity, it is crucial to build spaces of trust and respect where students can express their emotions, develop their socio-emotional skills, and promote better learning.

**Supplementary Materials:** The following are available online at https://www.mdpi.com/article/10.3390/socsci11070278/s1.

**Author Contributions:** Conceptualization, R.H.C.; methodology, R.H.C., L.S.T. and V.H.L.A.; formal analysis, R.H.C., M.E.R.S. and C.V.I.; investigation, R.H.C., L.S.T., V.H.L.A., M.E.R.S., C.V.I. and G.Á.G.; resources, G.Á.G.; data curation, R.H.C., L.S.T., V.H.L.A., M.E.R.S. and C.V.I.; writing—original draft preparation, review and editing, R.H.C.; project administration, R.H.C., L.S.T. and C.V.I. All authors have read and agreed to the published version of the manuscript.

**Funding:** This research was carried out with the support of the Instituto Politécnico Nacional and is part of the projects SIP20220731 and SIP20220847.

**Institutional Review Board Statement:** Not applicable.

**Informed Consent Statement:** Informed consent was obtained from all subjects involved in the study.

**Data Availability Statement:** We have included the document with the database.

**Acknowledgments:** Instituto Politécnico Nacional. Red de Seminarios Repensar. [Rethinking Seminar Network] and Margarita Pineda López.

**Conflicts of Interest:** The authors declare no conflict of interest.

### Appendix A. Instrument of Competency Evaluation

| Control questions: |
| --- |
| Gender (F) _____ (M) _____ |
| Age: ______________ |
| Educational level: High school _______ Bachelor _____ |
| AWARENESS |
| 1. I know my feelings/Conozco mis sentimientos |
| 2. I find it difficult to notice when my mood changes/Tengo dificultad para saber cuándo cambia mi estado de ánimo |
| 3. I find it difficult to differentiate my moods/Tengo dificultad para diferenciar mis estados de ánimo |
| 4. I know how to differentiate my feelings/Sé cómo diferenciar mis sentimientos |
| 5. I find it hard to realize what I am feeling/Me cuesta darme cuenta de lo que estoy sintiendo |
| 6. I have little connection with my feelings/Tengo poca conexión con mis sentimientos |
| 7. I find it difficult to recognize my emotions/Encuentro que es difícil reconocer mis emociones |
| 8. When I feel sad, I find it hard to know why/Cuando me siento triste es difícil saber por qué |
| REGULATION |
| 9. I can manage my emotions/Puedo manejar mis emociones |
| 10. It is difficult for me to act calmly when something makes me very nervous/Es difícil para mí actuar con calma cuando me siento nervioso |
| 11. I find it difficult to control my emotions/Es difícil controlar mis emociones |
| 12. I tend to lose control when something makes me angry/Tiendo a perder control cuando algo me hace enojar |
| 13. When I am angry, the worst of me comes out/Cuando estoy enojado, sale lo peor de mí |
| 14. When faced with a problem, I find it difficult to think clearly/Cuando enfrento problemas, me es difícil pensar con claridad |
| 15. When someone offends me, I am able to stay calm/Cuando alguien me ofende, puedo permanecer en calma |

| ASSERTIVENESS |
|---|
| 16. I am able to say the things that bother me/Puedo decir las cosas que me molestan |
| 17. I find it difficult to say that I disagree with something/Me es difícil decir cuando estoy en desacuerdo con algo |
| 18. It makes me very uncomfortable to say that something bothers me/Me siento muy incómodo al decir cosas que me molestan |
| 19. I say what I think even if others do not agree/Digo lo que pienso incluso si otros no están de acuerdo |
| 20. I find it easy to tell others what I think of them/Encuentro fácil decir a otros lo que pienso de ellos |
| 21. I find it easy to put limits on people when something bothers me/Encuentro fácil poner límites a la gente cuando algo me molesta |
| 22. I find it hard to set limits on people/Encuentro difícil poner límites a la gente |
| 23. I have a hard time saying "no"/Es difícil para mí decir "no" |
| 24. I express my opinions easily/Expreso mis opiniones fácilmente |
| 25. I get very nervous if I have to contradict someone/Me pongo nervioso si tengo que contradecir a alguien |
| 26. Even if you are right, I prefer to remain silent before arguing/Aunque tenga razón, prefiero callar que discutir |
| EXPRESSION |
| 27. I find it difficult to express my feelings towards others/Encuentro difícil expresar mis sentimientos hacia otros |
| 28. I find it easy to tell other people how much they are worth to me/Me resulta fácil decirle a otras personas lo que valen para mí |
| 29. I find it difficult to express what happens to me/Es difícil expresar lo que me pasa |
| 30. I can easily express what I am feeling/Puedo expresar fácilmente que estoy sintiendo |
| 31. People who know me say that I express myself well/Las personas que me conocen dicen que me expreso bien |
| 32. I am able to express my emotions when I talk to others/Puedo expresar mis emociones cuando hablo con otros |
| 33. I find it difficult to tell others how much they mean to me/Encuentro difícil decir a otros lo mucho que significan para mí |
| 34. I find it difficult to realize the feelings of others/Encuentro difícil darme cuenta de los sentimientos de otros |
| 35. I clearly say what happens to me, to others/Digo claramente lo que me pasa a mí, a los demás |
| OPTIMISM |
| 36. I notice when I am happy/Reconozco cuando me siento feliz |
| 37. I can focus on the positive aspects of life/Puedo enfocarme en los aspectos positivos de la vida |
| 38. Faced with difficult situations in life, I trust all will be well/Ante situaciones difíciles de la vida, confío en que todo saldrá bien |
| 39. When I set a goal, I accomplish it/Cuando me propongo un objetivo, lo cumplo |
| 40. I have a positive attitude towards life/Tengo una actitud positiva ante la vida |
| 41. I am able to see the bright side of things/Me gusta ver el lado positivo de las cosas |
| 42. I look to the future with hope/Veo el futuro con esperanza |
| EMPATHY |
| 43. Before criticizing a person, I try to think how I would feel if I were in their place/Antes de criticar a una persona, trato de pensar cómo me sentiría en su situación |
| 44. The problems of others affect me little/Los problemas de otros me afectan poco |
| 45. It is difficult for me to see things from another person's point of view/Es difícil para mí ver las cosas desde otro punto de vista |
| 46. When I am arguing, I try to put myself in the other person's position before saying something/Cuando discuto trato de ponerme en el lugar de otro antes de decir algo |
| 47. When I get angry with someone, I try to put myself in their place/Cuando estoy enojado con alguien, trato de ponerme en su lugar |
| SELF-EFFICACY |
| 48. I find it hard to enjoy life/Encuentro difícil disfrutar de la vida |
| 49. When I have a problem, it is difficult for me to solve it/Cuando tengo un problema, me resulta difícil resolverlo |
| 50. I doubt my ability to meet the goals I set for myself/Dudo de la habilidad de alcanzar mis propósitos por mi mismo |
| 51. I am good at solving the problems I have/Soy bueno resolviendo problemas |
| 52. I have little confidence in myself to achieve what I set out to do/Tengo poca confianza en mi mismo para lograr lo que me propongo |
| 53. I feel sure of myself in most situations/Me siento seguro de mí mismo en la mayoría de situaciones |
| 54. I am easily discouraged by the difficulties of life/Me desanimo fácilmente por las dificultades de la vida |
| 55. I have difficulties meeting the goals I set for myself/Tengo dificultades para definir mis propósitos |
| 56. I feel safe making decisions on my own/Me siento segura tomando decisiones por mí mismo |
| 57. I find it hard to think that things will turn out well/Me cuesta trabajo pensar que las cosas saldrán bien |
| 58. I think that things are easier for others than for me/Pienso que las cosas son más fáciles para otros que para mí |
| 59. I find it hard to finish what I start/Encuentro difícil terminar lo que empecé |
| 60. When I have many difficulties, it is difficult for me to think positive/Cuando yo tengo muchas dificultades, es difícil para mi pensar positivamente |
| 61. If there are complications, it is difficult for me to move forward/Si hay complicaciones me cuesta seguir para adelante |

| PRO-SOCIALITY |
|---|
| 62. I am willing to help others even when they do not ask me/Yo soy de ayudar a otros, aunque ellos no me lo pidan |
| 63. I am willing to help people who are in trouble/Yo soy de ayudar a las personas que tienen problemas |
| 64. I find it difficult to help other people/Yo encuentro difícil ayudar a otras personas |
| 65. I find it hard to accept that another person thinks differently/Encuentro difícil aceptar que otros piensen diferente |
| 66. I find it easier to do things that benefit me than others/Encuentro más fácil hacer cosas que me beneficien a mí que a otros |
| 67. When I know that something only benefits others, I hesitate to do it/Cuando sé que algo solo beneficia a otros, vacilo en hacerlo |
| AUTONOMY |
| 68. If I am determined about something, I do not let myself be influenced by others/Si estoy determinado a algo, no me dejo influenciar por otros |
| 69. I make important decisions without consulting others/Hago importantes decisiones sin consultar a otros |
| 70. They say that I am very dependent on my family/Ellos dicen que yo soy muy dependiente de mi familia |
| 71. I consult my family all the time/Consulto a mi familia todo el tiempo |
| 72. I depend on others to make decisions/Dependo de otros para tomar decisiones |

**Appendix B**

The plan of socialization activities carried out by students during the September–December 2021 semester (Huerta-Cuervo and Vicario 2021)

Objective: Create social meeting spaces in virtual environments to promote interpersonal relationships based on the exchange of ideas, collaboration, and companionship, aimed at strengthening socio-emotional competencies.

Competencies to develop:

1. Assume an empathic attitude towards group members.
2. Strengthen self-esteem and ability to relate to others.
3. Collaborate in the construction of answers and solutions to the questions and problems raised.
4. Apply receptive and expressive communication verbally and non-verbally in the dialogue and argumentation of ideas.
5. Practice values of respect, collaboration, tolerance, and dialogue.

Working method:

The meeting groups will be divided into teams of four or, at most, five students to discuss the topics listed below in separate rooms. Team members will be rotated in each session to promote the ability to maintain good relationships with other people.

During each session throughout the semester, 20 min will be allocated at the beginning of each class (from the second class) for students to carry out activities of integration, mutual knowledge, and socialization.

The sessions will have two types of activities: one with topics and tasks proposed by the teacher and the other with topics and tasks chosen by the students. For every three sessions with directed activities, a free session will be held, chosen, and organized by the students.

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
