# Peer review of "The Socio-Emotional Competencies of High School and College Students in the National Polytechnic Institute (Mexico)"

_socsci, doi:10.3390/socsci11070278_

Round 1

Reviewer 1 Report

In my opinion, the keywords must be rewritten to remove the compound ones and reduce them. Without a doubt, they must include the main conclusion of the work in the abstract.

In the introduction it is not clearly stated what the main objective of the work is, nor is it intuited what it is, just as this cannot be the objective (conduct a survey), this must be modified, and put a clear and understandable objective.

The figures are tables and they are hardly seen, they must be arranged according to the characteristics of the magazine, that is, all the figures must be changed and put in the format required by the edition. Research questions should be put in the form of hypotheses, H1,H2…..

Failure to do so leaves the results in the air, and therefore they cannot be taken for granted, they must state and list the research hypotheses, detail them since the work does not appear.

The discussion of results falls short and is not supported by the results of other studies, it must be extended and the results of the study compared with those used in the theoretical framework to support what has been done in the work.

Author Response

Dear Reviewer:

1.- The keywords were rewritten and it removed the compound ones and reduce them.

2.- We included the main conclusion of the work in the abstract.

3.- In the introduction was modified the objetive and put a clear and understandable objective.

4.- Research questions were be put in the form of hypotheses.

5.-  We changed the name of figures by tables and all the tables were changed and put in the format required by the edition.

6.-We precise the hypotheses and the analysis was made in relation with each one of the hypotheses.

7.- We expanded and supported the analysis in the results of other studies and compared with those used in the theoretical framework.

Thank you for your comments

Reviewer 2 Report

General:

-The paper has been organized as a thesis instead of a paper. Please modify the sections to become into a paper.

-Verify that your text was written in English.

Introduction:

  1. Please reorganize the introduction sections. You should include concepts, theories, or previous papers at the beginning.
  2. The hypothesis was separated into two sections (see lines 70 and 401). Please organize your introduction to establish hypotheses at the end of the introduction section.

Material and methods:

  1. Please include a table with the survey used in this study.
  2. Explain in detail the statistical analysis performed

Results:

  1. The” figures” are not figures, in addition, they have been written in Spanish, and the readers without knowledge of this language will be unable to understand the information. Please make figures or tables representing your results.
  2. Verify that the statistical result includes all data required to be reported.

Discussion

No comments.

Author Response

Dear reviewer:

Introduction

1.- We changed the document´s organization and we followed the instructions of the reviewer. 

2.- We rewrited all the paper in English.

3.- We reorganized the introduction sections and included concepts, theories, or previous papers at the beginning. We established the hypotheses at the end of the introduction section.

Material and methods:

4.-  We included a table with the survey used in this study and explained in detail the statistical analysis performed. 

Results

5.- We changed the name figures by tables and rewrited all the text in English.

6.- We included too all the data required to be reported

Thank you for your comments

Reviewer 3 Report

The research topic is topical, due to the consequences of the pandemic period on the development of socio-emotional competencies.
The hypotheses must be moved from the introductory part, after the theoretical part and before the methodological part. The hypotheses also appear in the Material and methods section. Must be removed from this section. They must be formulated only once.
The quoted sentences are too long.
Data analysis is formulated instead of the Analysis methodology.
The research instrument should be described in the Material and methods section.
All figures are not very clear.
I believe that the hypotheses should be reformulated, in accordance with the measured dimensions.

Author Response

Dear reviewer:

1.- We moved the hypotheses to the introductory part, after the theoretical part and before the methodological part.
2.- We removed the hypotheses from the section of material and methods.
3.-The quoted were reduced. We precised the metodology of analysis and included the description of instrument in the section of Material and methods.
4.- All the tables were changed by clear images.
5.- The hypotheses were reformulated in accordance with the measured dimensions.
Thank you for your comments

Round 2

Reviewer 2 Report

-Please avoid excessive use of questions in the manuscript.

-Provide the specific variables included in each statistical analysis.

-Provide the original version of the survey with English translation on the side as follows: 

  1. Conozco mis sentimientos ( I know my feelings)
  2. Me es dificil notar cuando tengo cambios de humor (I find it difficult to notice when my mood changes)

Author Response

Dear reviewed:

1.- We have eliminated some questions and we have replaced it by explanatory sentences 

2.- In each statistical analysis we have introduced the variables employed.

3.- We provide the original version of the survey like a support document, becasuse we employ excel. Like appendix we introduced the same survey with the  questions in English and Spanish, too, but in a compatible format with the template.

4. We check our English writing and ask for support in order to improve the document.

Thank you 

Best regards

Reviewer 3 Report

The hypotheses have been moved from the introductory part, after the theoretical part and before the methodological part. 
The research instrument has been describedin the Material and methods section.
All figures are very clear.

Author Response

Dear reviewer

Thank you for your comments

Best regards
